# Validating the Arabic version of the Connor-Davidson resilience scale among university students

**Othman A. Alfuqaha** [ORCID] *

Counseling and Mental Health Department, Faculty of Educational Sciences, The World Islamic Sciences & Education University W.I.S.E, Amman, Jordan

* Othman.fuqaha@wise.edu.jo

## Abstract

### Background

The Connor-Davidson Resilience Scale (CD-RISC) stands out as a widely used measure of psychological resilience. The original CD-RISC consists of 25 items, commonly referred to as CD-RISC-25.

### Aim

This study aims to validate the Arabic version of the CD-RISC-25 involving a sample size of 1220 participants from three universities in Jordan.

### Methods

The researcher conducted a methodological investigation to examine the validation process. This included an examination of the translation process and an evaluation of content validity, which was assessed using the content validity index (CVI). Construct validity was assessed through exploratory factor analysis and confirmatory factor analysis, while convergent and discriminant validity were both evaluated using composite reliability (CR) and the square root of the average variance extracted.

### Results

Results showed a prominent level of psychological resilience 76.74±15.2 among the participating university students. Regarding the validity of the Arabic version of CD-RISC-25, the CVI yielded a value of 0.88, signifying a robust level of content validity. The analysis identified four constructs that accounted for 56.26% of the total variance. The goodness-of-fit indices, including goodness of fit index of 0.91, comparative fit index of 0.91, incremental fit index of 0.93, Tucker-Lewis index of 0.94, and root mean square error of approximation of 0.048, provided strong evidence supporting the alignment between the observed data and the hypothesized constructs. Discriminant and convergent validity were found to support the validity of the Arabic version of CD-RISC-25. Regarding the reliability, it demonstrated

**Data Availability Statement:** Data cannot be shared publicly because of the regulations of The World Islamic Sciences and Education University (W.I.S.E). Data are available from the WISE Ethics Committee at the faculty of educational sciences

(contact via deewan.educational@wise.edu.jo) for researchers who meet the criteria for access to confidential data.

**Funding:** The author(s) received no specific funding for this work.

**Competing interests:** The authors have declared that no competing interests exist.

excellent reliability, with a total Cronbach's alpha of 0.934 and all CR values surpassing the threshold of 0.70, thereby further establishing its overall robustness.

## Conclusion

The results provide substantial evidence for the validity and reliability of the translated Arabic CD-RISC-25.

## Introduction

Psychological resilience is a crucial construct in psychology that refers to an individual's ability to adapt and thrive in the face of adversity. It has been recognized as a protective factor against various mental health problems, including anxiety, depression, and posttraumatic stress disorder (PTSD) [1]. Resilience can be defined as the capacity of individuals to confront, adapt, and efficiently recover from diverse stressors encountered throughout their lives [2]. To measure psychological resilience, several scales have been developed, including brief resilient coping scale [3], response to stressful events scale [4], and hardiness scale [5]. Of these scales, the Connor-Davidson resilience scale (CD-RISC) stands out as a widely used measure of psychological resilience that has been validated across various cultures and languages [6, 7]. The original CD-RISC consists of 25 items, commonly referred to as CD-RISC-25, which are systematically allocated into five dimensions namely, positive acceptance and relationships (5 items), trust in one's ability (7 items), competence (8 items), control (3 items), and spirituality (2 items) [1]. Two abbreviated scales comprising 10 and 3 items respectively have been derived from the original CD-RISC-25 to condense the number of items while retaining its essence [8]. However, it is important to note that these abridged scales have not gained extensive traction across various contexts, and they lack the specific multidimensional framework exhibited by the original CD-RISC-25. In contrast, the CD-RISC-25 has demonstrated its robustness by undergoing assessment across diverse populations, including individuals with mental health conditions [1], university students [9], survivors of trauma [10], and individuals with opioid dependency [11].

University students often experience profound burnout, considerable life stressors, additional responsibilities, and a host of challenges throughout their academic journey. Numerous studies have highlighted the unique pressures that university students encounter, including academic demands, extreme stressors, financial constraints, and personal growth expectations [12, 13]. Furthermore, it was observed that Jordanian university students exhibited pronounced signs and symptoms of PTSD following a crisis situation [14] as well as general population following the COVID-19 pandemic [15]. Given the significant impact of these stressors and crises on student well-being, validating a psychological resilience scale among Jordanian university students is of paramount importance to the wider Jordanian population. Despite the growing interest in psychological resilience study in Arab countries, there is a scarcity of validated psychological resilience measures that are culturally and linguistically appropriate for this population. While there have been some studies that have investigated the psychometric properties of the Arabic version of the CD-RISC-25 in Arab populations, the available evidence is limited. For instance, only one study has explored the psychometric properties of the Arabic version of the CD-RISC-25 in a sample of Lebanese women [16]. However, the sample size in this study was small, consisting of only 63 pregnant women. In contrast, a recent study focused only on validating the CD-RISC-10 in the Arabic language [17]. Hence, there exists a requirement to validate the CD-RISC-25 within a more extensive demographic, such as

university students, who frequently encounter significant stressors within their academic pursuits. To our knowledge, there have been no prior Arabic studies that have examined the validity and reliability of the CD-RISC-25 specifically among university students. Additionally, the validation of the CD-RISC-25 among university student offers valuable insights for researchers and students in comprehending coping skills concerning various stressors.

## Aim

The primary aim of this study is to validate the Arabic version of the CD-RISC-25 using a sample of university students from Jordan as well as to explore the level of psychological resilience among them. Specifically, the researcher employs various validity and reliability measures to explore the underlying factor structure of the CD-RISC-25 within this specific population.

# Methods and materials

## Participants and settings

The present study was carried out among a cohort of Jordanian university students who were at least 18 years of age, and were enrolled in major schools of medical, scientific, and humanities. University students who did not meet these criteria, including college students with a history of mental illness, and those who were not willing to participate, were excluded from the study. To solicit participation from potential participants, an official letter was sent to all universities in Jordan. Ultimately, three universities, comprising two public and one private universities, provided agreement letters indicating their willingness to participate. In order to ensure a sufficiently robust sample size, it is recommended to include a participant pool that ranges from 10 to 15 times the number of items in the measurement instrument. Considering the CD-RISC encompasses 25 items, the requisite number of participants per university would be 375 students [18]. Thus, the researcher recruited a total of 500 university students from each of the three participating universities, thereby representing the study sample. Between the months of January and March 2023, the researcher administered paper-based surveys at the participating universities to obtain data from the study sample. Surveys containing incomplete or missing information were excluded from the final dataset, resulting in the inclusion of 788 valid surveys from the two public universities and 432 valid surveys from the private university. The total number of completed surveys obtained from the three universities was 1220 out of 1500, resulting in a response rate of 81.3%. Prior to participation, participants were informed about the study's objectives and procedures, and were provided with the option to withdraw from the study at any point without facing any repercussions. The survey took approximately 5–10 minutes to complete.

In the data analysis phase, this research relies upon the utilization of two distinct software programs, namely SPSS version 23 and Analysis of Moment Structures (AMOS) software version 25. SPSS is employed for the computation of fundamental statistical measures such as frequencies, means, standard deviations, explanatory factor analysis (EFA), and the Cronbach's alpha coefficient. On the other hand, the AMOS software is employed to conduct a comprehensive examination of the interrelationships among items, particularly during the confirmatory factor analysis (CFA) stage.

## The original CD-RISC translation process

The translation process of the CD-RISC-25 into Arabic language involves several steps. In this study, as a previous study Bizri et al. [16] had already translated the scale into the Arabic language, the process of forward translation and back translation was skipped. We obtain

permission from the original authors to make any required modifications to the CD-RISC-25. Next, a comparison was conducted between the original English version of the CD-RISC-25 and the translated Arabic version as translated by a previous study. After that, the original and translated versions of the CD-RISC-25 in Arabic language were presented to experts in the relevant fields, who were asked to provide their opinions and suggest any necessary amendments to the items. However, participants were asked to respond to each item on the translated Arabic CD-RISC-25 using a 5-point Likert-type scale, ranging from 0 "Never" to 4 "Always." In order to control response bias, negative items were reversed. The score ranges from 0 to 100, where higher scores correspond with higher resilience.

## Validity

**Content validity.** The researcher employed two approaches: obtaining feedback from a representative sample and seeking evaluation from experts in the field. For the sample assessment, we engaged 30 university students who were asked to evaluate the clarity, appropriateness, and comprehensibility of the items in the Arabic version of the CD-RISC-25. Concurrently, we sought the expertise of eight professionals holding PhDs in the fields of counseling, psychology, nursing, and mental health to further enhance the instrument's relevance and appropriateness of the Arabic version of CD-RISC-25. By using three point Likert scale (clear, unclear, I don't know), we facilitated the assessment of content validity index (CVI) by calculating the experts' average rating on the CD-RISC-25 items [19].

**Construct validity.** The researcher conducted both EFA and CFA using recommended measures including intercorrelation varimax ($> 0.30$), eigenvalues ($>1$), Kiser-Meller-Olken (KMO$> 0.70$), and Bartlett's test of sphericity (P$< 0.05$) for the EFA. For the CFA, factor loadings ($> 0.30$), correlations between constructs less than unity, and fit indices including Chi-square test ($x$), goodness of fit index (GFI $>0.90$), comparative fit index (CFI $>0.90$), incremental fit index (IFI $>0.90$), Tucker-Lewis Index (TLI $>0.90$), and root mean square error of approximation (RMSEA $<0.05$) were used as criteria [20].

**Discriminant and convergent validity.** The researcher employed composite reliability (CR) and the square root of the average variance extracted (AVE). The CR values exceeding 0.70 indicate strong convergent validity, while the square root of the AVE demonstrated greater values compared to the inter-factor correlations of CD-RISC-25 [20].

## Reliability

The researcher performed both Cronbach's alpha and composite reliability (CR) to assess the reliability of CD-RISC-25. A Cronbach's alpha coefficient surpassing 0.70 signifies enhanced internal consistency [21], while CR values exceeding 0.70 are indicative of heightened reliability [22].

## Ethical guidelines

The researcher ensured that ethical clearance was obtained prior to data collection, and that all ethical guidelines were strictly followed throughout the study. Informed consent was obtained from participants, and no identifying information was requested to ensure confidentiality. The first paper in our survey presents the primary aim of this study, and at the end of the page, participants were asked to provide their name and signature as a sign of agreement to participate. The study was approved by the research ethics committee at the counseling and mental health department, faculty of education sciences, The World Islamic Sciences & Education University (No: 9/1/2/8366).

## Results

### Demographic information and CD-RISC score

A total of 1220 university students completed a valid survey. Most of them were female (76.5%), single (92.6%) and more than half of them were between the age of 18 and 20 years old (56.5%). Participants students in public university (64.6%) and humanistic schools (56.5%) were more responded to our surveys. The freshman academic year (25%) and scientific schools (10.8%) were the least percentage of participants as illustrated in Table 1.

Table 1 also presents the overall mean resilience score among university students, which was 76.74±15.2, indicating a high level of resilience. For all dimensions of the translated CD-RISC-25, perceived levels were consistently high.

### Translation process

The assessment involved a comprehensive evaluation by experts, focusing on the comparative analysis between the translated Arabic version and the original English version of CD-RISC-25. This scrutiny was directed toward ensuring both semantic congruence and cultural resonance. Thus, no items were subjected to deletion in the Arabic version; however, amendments

**Table 1. Demographic information and resilience score (N = 1220).**

| Demographic information | Descriptive | n (%) | M±SD |
|---|---|---|---|
| Gender | Male | 287 (23.5%) | |
| | Female | 933 (76.5%) | |
| Marital status | Single | 1129 (92.6%) | |
| | Married | 71 (5.7%) | |
| | Other | 20 (1.6%) | |
| Age (years) | 18–20 | 689 (56.5%) | 20.3±3.96 |
| | >20 | 531 (43.5%) | |
| Grade | Approval | 157 (12.8%) | 2.70±0.95 |
| | Adequate | 326 (26.7%) | |
| | Very good | 469 (38.5%) | |
| | Outstanding | 268 (22%) | |
| Type of university | Public | 788 (64.6%) | |
| | Private | 432 (35.4%) | |
| Type of school | Health | 398 (32.6%) | |
| | Scientific | 133 (10.8%) | |
| | Humanistic | 689 (56.5%) | |
| Academic year | Freshman | 305 (25%) | 2.37±1.07 |
| | Sophomore | 407 (33.4%) | |
| | Junior | 254 (20.8%) | |
| | Senior or above | 254 (20.8%) | |
| | | Min-Max | |
| Resilience | Total resilience score (25-items) | 0–100 | 76.74±15.2 |
| | Secure relationship (5-items) | 0–20 | 15.8±3.4 |
| | Trust (7-items) | 0–28 | 22.4±4.5 |
| | Competence (8-items) | 0–32 | 22.6±5.6 |
| | Control (5-items) | 0–20 | 15.6±3.8 |

**Notes**: n (%): Number (Percentage), M±SD: Mean ± Standard Deviation, Min-Max: minimum-Maximum.

in some words were incorporated into the Arabic version of the items, attuning them to contextual Arabic cultural significance.

## Validity

**Content validity.** During the assessment of content validity, the expertise of eight professionals demonstrated minor modifications and clarifications to the item in terms of Arabic words. Following their modifications and recommendations, no items were removed from the Arabic version of CD-RISC-25. The CVI yielded a value of 0.88, signifying a robust level of content validity. Additionally, it was noted during the pilot study that some students found certain items to be unclear in the Arabic version. As a result, modifications were made to those items to enhance clarity.

**Construct validity.** In the EFA, the methodology involved several steps. First, the KMO measure was computed, yielding a result of 0.96. Bartlett test of sphericity was conducted, producing ($x = 14170.42$) with 300 degrees of freedom and $p < 0.001$. Furthermore, the intercorrelation varimax method was performed, revealing scores above 0.30 for the factors, as illustrated in Table 2. As indicated in Fig 1, constructs with eigenvalues greater than 1 are

**Table 2. Factor loading of the translated Arabic CD-RISC-25 (n = 1220).**

| Items | Factor 1 | Factor 2 | Factor 3 | Factor 4 |
|---|---|---|---|---|
| | Secure relationship | Trust | Competence | Control |
| Item 1 | 0.44 | | | |
| Item 2 | 0.76 | | | |
| Item 3 | 0.40 | | | |
| Item 4 | 0.61 | | | |
| Item 5 | 0.63 | | | |
| Item 6 | | 0.44 | | |
| Item 7 | | 0.57 | | |
| Item 8 | | 0.54 | | |
| Item 9 | | 0.47 | | |
| Item 10 | | 0.66 | | |
| Item 11 | | 0.56 | | |
| Item 12 | | 0.73 | | |
| Item 13 | | | 0.54 | |
| Item 14 | | | 0.62 | |
| Item 15 | | | 0.59 | |
| Item 16 | | | 0.42 | |
| Item 17 | | | 0.49 | |
| Item 18 | | | 0.63 | |
| Item 19 | | | 0.73 | |
| Item 20 | | | 0.75 | |
| Item 21 | | | | 0.67 |
| Item 22 | | | | 0.77 |
| Item 23 | | | | 0.71 |
| Item 24 | | | | 0.73 |
| Item 25 | | | | 0.55 |
| **Initial eigenvalues** | 10.27 | 1.69 | 1.09 | 1.02 |
| **Percentages of variance explained** | 41.08 | 6.75 | 4.36 | 4.08 |
| **Cumulative variance** | 41.08 | 47.83 | 52.19 | 56.26 |

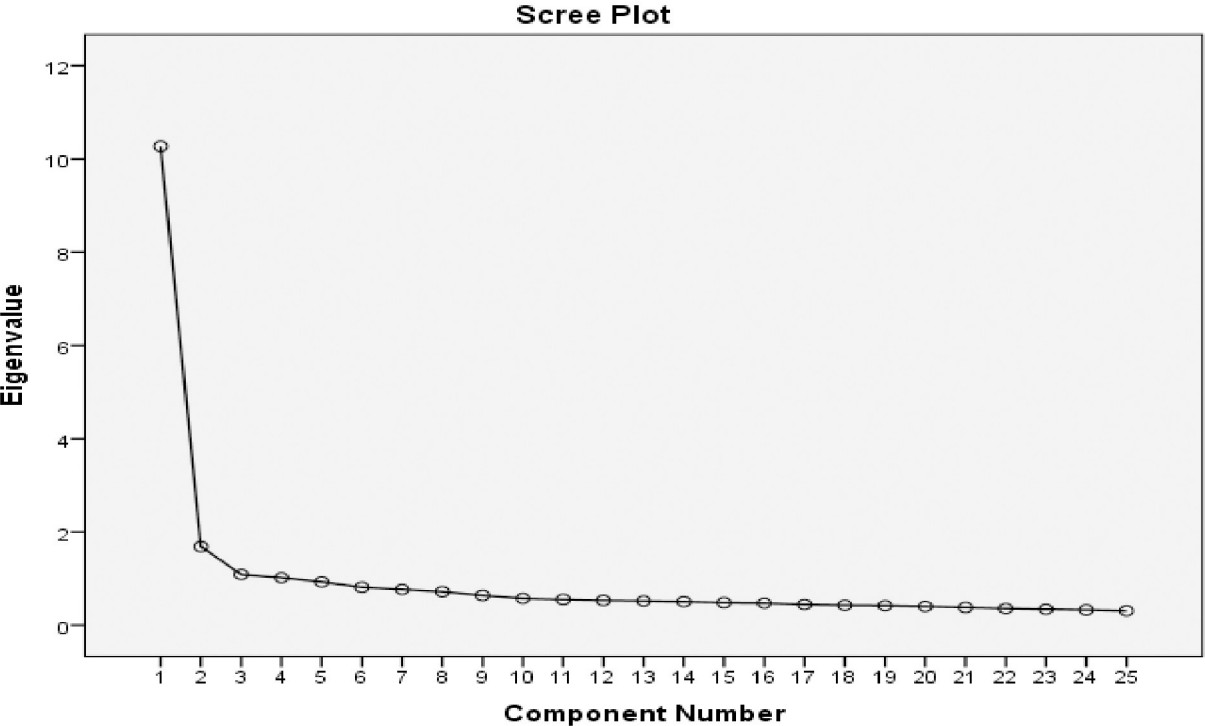

**Fig 1. Scree plot of the translated Arabic CD-RISC-25.**

considered constructs. In this regard, results identified four constructs namely secure relation-ship (5-items), trust (7-items), competence (8-items), and control (5-items) with eigenvalues exceeding 1. Lastly, the cumulative loading of the four constructs accounted for 56.26% of the total variance, indicating their substantial explanatory power in relation to the observed data (Fig 1).

The 4-factor model in the CFA demonstrated factor loading values ranging from 0.48 to 0.91. The statistical significance of the model was supported by chi-square value below 0.001. The goodness-of-fit indices, including GFI (0.91), CFI (0.91), IFI (0.93), TLI (0.94), and RMSEA (0.048), indicated a favorable fit between the observed data and the hypothesized constructs.

**Discriminant and convergent validity.** The researcher conducted calculations of com-posite reliability (CR) for the four constructs of the Arabic version of CD-RISC-25. Results revealed that all CR values exceeded the threshold of 0.70, indicating strong convergent valid-ity for the Arabic version of CD-RISC-25. Furthermore, upon examining the square root of the average variance extracted (AVE), results found that all values were higher than the inter-correlations between the constructs, as indicated in Table 3. These findings provide evidence supporting both discriminant and convergent validity of the Arabic CD-RISC-25.

## Reliability

The translated Arabic version of CD-RISC-25 exhibited a total Cronbach's alpha of 0.934, indi-cating high internal consistency. The factors within the scale ranged from 0.74 to 0.88, further supporting its internal reliability as displayed in Table 3. Likewise, all CR values exceeded 0.70, thereby demonstrating that the scale meets another crucial criterion for reliability, thus further enhancing its overall reliability.

**Table 3. The reliability indicators of the translated Arabic CD-RISC-25 (n = 1220).**

| Constructs | Secure relationship | Trust | Competence | Control | Cronbach's alpha | CR | AVE |
|---|---|---|---|---|---|---|---|
| Secure relationship | **0.58** | | | | 0.74 | 0.71 | 0.34 |
| Trust | 0.32 | **0.46** | | | 0.77 | 0.82 | 0.21 |
| Competence | 0.35 | 0.33 | **0.43** | | 0.88 | 0.87 | 0.18 |
| Control | 0.40 | 0.37 | 0.31 | **0.62** | 0.74 | 0.82 | 0.47 |

*Note.* CR: Composite reliability. AVE: Average variance extracted, Diagonal line: The square root of AVE.

## Discussion

This study underscores the importance of resilience among university students. The sample consisted of approximately two-thirds females, and the average age of participating students was 20.3±3.96. The findings revealed that a majority of students demonstrated noteworthy psychological resilience when confronted with challenging situations throughout their academic journey. Moreover, the result showed that the Arabic version of CD-RISC-25 is valid and reliable among university students. This result is consistent with similar studies conducted in other languages, such as German [23], Portuguese [10], and Thai [24]. The translated CD-RISC-25 into the Arabic language provides several benefits for university students by exploring their ability to adapt with various stressors.

The translation process of the original CD-RISC-25 into the Arabic language, as conducted by Bizri et al. [16], appears to be adequate with a few modifications needed for certain items regarding linguistics. However, the assessment of content validity involved the collaboration of eight qualified professionals with expertise in counseling, psychology, nursing, and mental health. Their valuable input and recommendations were incorporated into the Arabic version of CD-RISC-25 to enhance its relevance and appropriateness. Notably, no items were removed from the scale, indicating that the modifications made based on expert feedback successfully maintained the integrity of the instrument. Furthermore, adjustments were made to specific items following the pilot study, addressing concerns about clarity and ensuring improved comprehension.

Nevertheless, the researcher employed both EFA and CFA to obtain construct validity. The KMO measure yielded a high value of 0.96, indicating the adequacy of the data for factor analysis. Additionally, the Bartlett test of sphericity demonstrated statistical significance, affirming the suitability of EFA. The intercorrelation varimax method revealed factor scores above 0.30, providing evidence for distinct factors. Moreover, the cumulative loading of the four constructs namely secure relationship, trust, competence, and control accounted for a substantial portion (56.26%) of the total variance, indicating their strong explanatory power in relation to the observed data. The original CD-RISC-25 comprises five constructs, whereas our validated process includes four constructs. This result has engendered debate in previous studies, with some findings corroborating our observations, as seen in the study of Wu et al. [25]. Conversely, other studies have proposed alternative models for the CD-RISC-25. These alternatives encompass a 2-factor model among United States military veterans [26], a 3-factor model among the Chinese military [27], and a 5-factor model among Iranian girls [28]. In the CFA, the 4-factor model of the Arabic CD-RISC-25 exhibited favorable factor loading values ranging from 0.48 to 0.91, further supporting its construct validity. Additionally, the goodness-of-fit indices indicated a favorable fit between the observed data and the hypothesized constructs. These findings demonstrate that the proposed model accurately captures the relationships among the variables and provides a valid framework for assessing resilience among university students.

Discriminant and convergent validity of the Arabic CD-RISC-25 were found to support the validity of the Arabic version of CD-RISC-25 among university students. This particular study contradicts the findings of a recent Dutch study, which concluded that the CD-RISC-25 did not demonstrate convergent validity, while the CD-RISC-10 in the same study did exhibit convergent validity [29]. For the internal consistency, as assessed through Cronbach's alpha and CR, the scale exhibited satisfactory alpha values 0.934 and provided further evidence of the scale's overall reliability. The higher Cronbach alpha value may be attributed to both the larger number of participating students and the consistency observed among the items of the Arabic version of CD-RISC-25. These findings affirm the consistency and stability of measurements, enhancing the overall reliability of the translated Arabic CD-RISC-25.

Various scales are frequently used to evaluate resilience, including academic resilience scale [30], protective factors scale [31], 6-factor predictive resilience scale [32], and ego resilience scale [33]. These scales have demonstrated their validity and reliability across diverse populations and professions. Despite a prevailing trend favoring the CD-RISC-10, the researcher holds the opinion that the CD-RISC-25 offers a more comprehensive evaluation of resilience, encompassing a wider spectrum of resilience indicators such as the understanding of stressors, adaptive capacity, and the potential for personal growth. However, it is important to acknowledge certain limitations that may affect generalizability. One limitation is the reliance on a specific sample of university students, which may not fully represent the broader population. Furthermore, the overrepresentation of female university students within our study's sample could potentially limit the generalizability of our findings. Utilizing surveys is susceptible to response bias, which should be regarded as an additional limitation in this study. However, future research should consider utilizing diverse samples and conducting cross-cultural validations to enhance the instrument's applicability across different populations.

## Conclusion and recommendations

The results of this study provide substantial evidence for the validity and reliability of the translated Arabic CD-RISC-25. The thorough assessment of content, construct, discriminant, and convergent validity, as well as internal consistency and composite reliability, confirm the robustness and applicability of the instrument in assessing resilience among Arabic-speaking university students. These findings contribute to the existing literature and offer researchers and practitioners a validated tool for resilience assessment in this population. It is recommended to replicate the study with diverse samples, conduct longitudinal study, perform cross-cultural validation, analyze item-level performance, examine external criterion validity, investigate sensitivity to intervention, and validate the instrument in specific subpopulations. These recommendations will contribute to refining the Arabic CD-RISC-25 and enhancing its applicability in research and practice.

## Author Contributions

**Conceptualization:** Othman A. Alfuqaha.

**Data curation:** Othman A. Alfuqaha.

**Formal analysis:** Othman A. Alfuqaha.

**Funding acquisition:** Othman A. Alfuqaha.

**Investigation:** Othman A. Alfuqaha.

**Methodology:** Othman A. Alfuqaha.

**Project administration:** Othman A. Alfuqaha.

**Resources:** Othman A. Alfuqaha.

**Software:** Othman A. Alfuqaha.

**Supervision:** Othman A. Alfuqaha.

**Validation:** Othman A. Alfuqaha.

**Visualization:** Othman A. Alfuqaha.

**Writing – original draft:** Othman A. Alfuqaha.

**Writing – review & editing:** Othman A. Alfuqaha.

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
