## [Decision Letter · Decision Letter 0]

21 Aug 2023

PONE-D-23-22861Validating the Arabic Version of the Connor-Davidson Resilience Scale Among University StudentsPLOS ONE

Dear Dr. Alfuqaha,

Thank you for submitting your manuscript to PLOS ONE. After careful consideration, we feel that it has merit but does not fully meet PLOS ONE’s publication criteria as it currently stands. Therefore, we invite you to submit a revised version of the manuscript that addresses the points raised during the review process.

Dear editor: Thanks for working on validating a required tool, however, you need to:You need to work on the reviewers feedback to get your paper being accepted Although one reviewer suggested to reject the paper, I do agree with second reviewer's point view which to do a major modificationsFurther details is needed about the scale and other similar scales. why your target population were students.how many participants included in the pilot study? The expert and CVI need further details. ==============================

We look forward to receiving your revised manuscript.

Kind regards,

Fadwa Alhalaiqa

Academic Editor

PLOS ONE

Journal Requirements:

4. We note you have included a table to which you do not refer in the text of your manuscript. Please ensure that you refer to Table 1 in your text; if accepted, production will need this reference to link the reader to the Table.

Additional Editor Comments:

Dear Authors

The paper validated a required tool.

But why you selected students as your target group need to be explained.

You need to determine the background of experts who did the CVI, give more details about CVI.

You need to consider the reviewers feedback

Good luck

Reviewers' comments:

Reviewer's Responses to Questions

**Comments to the Author**

1. Is the manuscript technically sound, and do the data support the conclusions?

Reviewer #1: Partly

Reviewer #2: Yes

2. Has the statistical analysis been performed appropriately and rigorously? 

Reviewer #1: No

Reviewer #2: Yes

3. Have the authors made all data underlying the findings in their manuscript fully available?

Reviewer #1: Yes

Reviewer #2: No

4. Is the manuscript presented in an intelligible fashion and written in standard English?

Reviewer #1: No

Reviewer #2: Yes

5. Review Comments to the Author

Reviewer #1: ID: PONE-D-23-22861

Title: Validating the Arabic Version of the Connor-Davidson Resilience Scale Among

University Students

Thank you for providing a chance to review this manuscript.

Detailed information:

Abstract

Overall: 1) “The Connor-Davidson Resilience Scale (CD-RISC-25)”, What does "25" in parentheses mean? This kind of detail also needs to be explained clearly. 2) The abstract should be divided into background, purpose, research methods, results, and conclusions for writing. This way, readers can quickly and roughly understand your overall research. Do not mix methods, results, and conclusions together. 3) The results section should list all key values or specific statistical results(value), including GFI, CFI, IFI, TLI, RMSEA, etc. 4) The statistical methods for verifying reliability and validity should also be written out.

Introduction

Paragraph 1, page 3: 1) The definition and role of mental resilience can be explained in more detail. 2) What items are included in the Connor-Davidson resilience scale (CD-RISC)? What is the scoring method? How about the original version and reliability? What are its main uses? 3) The difference between CD-RISC-25 and the other two scales is not clear enough, and the reason why this scale was chosen instead of the other two scales is not clearly written.

Paragraph 2, page 4: Where is it important? The significance of this research is too simplistic. Please clarify the purpose and significance more clearly. Overall, this paragraph is too abrupt. Adding some background on mental health issues among college students in Jordan would help illustrate the purpose of your research.

Methods and materials

Aim, page 4: Please place this paragraph at the end of the introduction section.

Design, page 4: The first sentence is repetitive and meaningless; Too concise, no actual content.

Participants and settings, page 5: 1) The inclusion and exclusion criteria should be more clearly described. For example, college students with a history of mental illness should be excluded. 2) What is meant to ensure sufficient sample size? How is the sufficient sample size calculated? This important content should be written out. 3) How to deal with missing values and quality control?

The original CD-RISC translation process, page 5: What is the total score? What do high and low scores represent? After the experts provided their opinions, what modifications have you made to the scale?

Reliability, page 7: Please clearly state the method used for reliability verification.

Results

Demographic information

Table 1: 1) With such a large gender gap, how can the results be representative? 2) Make sure that the percentages in each category add up to 100%. 3) It is recommended to list the scores of each category (Mean±SD) in the table 1.

Discussion

As suggested above, first of all, you should conduct a brief descriptive analysis to describe the basic scores and conditions of psychological resilience of Jordanian college students.

First of all, your research objects are college students from three schools in Jordan. I think this group is not representative, and the article does not explain why college students are chosen for the research. Second, there have been previous studies. Thirdly, your study is only cross-sectional and cannot elaborate the measurement invariance of the scale in depth. Finally the innovation of this study is not enough, and many details were not paid attention to, making the detailed description too brief.

Thank you and my best,

Your reviewer

Reviewer #2: Thank you for submitting this manuscript which is both timely and contemporary. Before this manuscript can be considered for publication, there are few points which the author should revisit.

1. The excluding criteria includes “high school students”. Not sure if this should be included as excluding criterion as the university student cannot have only high school students. Please reconsider.

2. Although the sample size of 1220 seems to be adequate, there is no scientific or empirical justification or mentioning of the “rule of thumb” which are widely used in the literature to justify this sample size. Please consult the following key references which may give a better explanation in relation to the number of participants per items:

• Spielberger C, Gorusch R, Lushene R, Vagg P, Jacobs G. State trait anxiety for adults. Mind Garden, Redwood city, Calif, USA. 1983.

• 15.Pedhazur E. Multiple regression in behavioral research, Orlando, FL: Harcourt Brace. 1997.

• 16.Kline RB. Convergence of structural equation modeling and multilevel modeling: na; 2011.

• 17.Field A. Discovering statistics using IBM SPSS statistics: sage; 2013.

3. Most of the participants were female (i.e., 77%). Does this have an impact on the participants responses?

4. The Cronbach alpha for the sample was excellent (i.e. more than 0.9)., but the author should also acknowledge that as the sample size increases, the Cronbach alpha will also increase and this measure become less credible indicator for the internal consistency of the scale.

5. In p. 9. It was mentioned that some changes to the Arabic version of the scale were implemented following the pilot study. Please clarify these changes and why they were implemented?

6. It was clear from the scree plot that a two factors model was probably the best, yet the author has opt to use four factors model. Please provide an empirical justification for this decision.

7. In the limitation section, the author suggest that the “ Connor-Davidson Resilience Scale 15 the study focused solely on the Arabic-speaking context, limiting the generalizability to other cultural or linguistic groups “. This is an incorrectly paraphrased sentence. The whole paper is focused on the Arabic version of the scale, so why would the author question the fact that this study was conducted on Arabic-speaking sample?

6. PLOS authors have the option to publish the peer review history of their article (what does this mean?). If published, this will include your full peer review and any attached files.

Reviewer #1: No

Reviewer #2: **Yes: **Dr. Mansour Mansour. Associate Professor and Senior Fellow of High Education Academy

Fatima College for Health Sciences – Ajman Campus

---

## [Author Response · Author response to Decision Letter 0]

3 Sep 2023

Response to Journal and Reviewers comment

Response to the journal requirements

We appreciate your careful review of our manuscript and thank you for the opportunity to submit our revised manuscript and the point-by-point responses to the journal requirements. We hope that we have satisfactorily addressed all issues raised by the journal requirements.

Comment

Response 1: Done

Response 2: Done

Response 3: Data cannot be shared publicly because of the regulations of The World Islamic Sciences and Education University (W.I.S.E). Data are available from the WISE Ethics Committee at the faculty of educational sciences (contact via deewan.educational@wise.edu.jo) for researchers who meet the criteria for access to confidential data.

Response a: We added a new paragraph in our cover letter that “Data cannot be shared publicly because of the regulations of The World Islamic Sciences and Education University (W.I.S.E). Data are available from the WISE Ethics Committee at the faculty of educational sciences (contact via deewan.educational@wise.edu.jo) for researchers who meet the criteria for access to confidential data”.

Response b: Please see the above response a.

4. We note you have included a table to which you do not refer in the text of your manuscript. Please ensure that you refer to Table 1 in your text; if accepted, production will need this reference to link the reader to the Table.

Response 4:

As suggested, Table 1 was added in the text.

Dear editor,

We would like to thank you and the reviewer for their valuable comments and feedbacks. Point-by-point responses to reviewers are listed below.

Editor comments:

Dear editor: Thanks for working on validating a required tool, however, you need to:

You need to work on the reviewers feedback to get your paper being accepted.

Response: Thanks.

Although one reviewer suggested to reject the paper, I do agree with second reviewer's point view which to do a major modifications.

Response: Thanks

Further details is needed about the scale and other similar scales. 

Response: Added at the end of discussion part as follows: Various scales are frequently employed to evaluate resilience, including Academic Resilience Scale [26], Protective Factors Scale [27], 6-Factor Predictive Resilience Scale [28], and Ego Resilience Scale [29]. These scales have demonstrated their validity and reliability across diverse populations and professions.

why your target population were students.

Response: We added a new paragraph illustrating why we chose this group as follows: Our decision to concentrate on university students was driven by the recognition that this group often experiences profound burnout, considerable life stressors, additional responsibilities, and a host of challenges throughout their academic journey. Numerous studies have highlighted the unique pressures that university students encounter, including academic demands, financial constraints, and personal growth expectations [13, 14]. 

how many participants included in the pilot study? 

Response: 30 participants. Please see the content validity under the method section.

The expert and CVI need further details.

Response: More information was added regarding the experts and CVI. Please see the content validity in our revised manuscript in method and results sections.

Reviewer comments

Reviewer #1: 

ID: PONE-D-23-22861

Title: Validating the Arabic Version of the Connor-Davidson Resilience Scale Among University Students

Thank you for providing a chance to review this manuscript.

Detailed information:

Comment 1:

Abstract

Overall: 1) “The Connor-Davidson Resilience Scale (CD-RISC-25)”, What does "25" in parentheses mean? This kind of detail also needs to be explained clearly. 2) The abstract should be divided into background, purpose, research methods, results, and conclusions for writing. This way, readers can quickly and roughly understand your overall research. Do not mix methods, results, and conclusions together. 3) The results section should list all key values or specific statistical results(value), including GFI, CFI, IFI, TLI, RMSEA, etc. 4) The statistical methods for verifying reliability and validity should also be written out.

Response 1:

Thank you for your valuable comments. All of your concerns have been addressed and incorporated in accordance with your suggestions. Please see our revised abstract section. 

Comment 2:

Introduction

Paragraph 1, page 3: 1) The definition and role of mental resilience can be explained in more detail. 2) What items are included in the Connor-Davidson resilience scale (CD-RISC)? What is the scoring method? How about the original version and reliability? What are its main uses? 3) The difference between CD-RISC-25 and the other two scales is not clear enough, and the reason why this scale was chosen instead of the other two scales is not clearly written.

Paragraph 2, page 4: Where is it important? The significance of this research is too simplistic. Please clarify the purpose and significance more clearly. Overall, this paragraph is too abrupt. Adding some background on mental health issues among college students in Jordan would help illustrate the purpose of your research.

Response 2:

Thank you for your valuable comments. All of your concerns have been addressed and incorporated in accordance with your suggestions. Please see our revised introduction section. 

Comment 3:

Methods and materials

Aim, page 4: Please place this paragraph at the end of the introduction section. Done.

Design, page 4: The first sentence is repetitive and meaningless; Too concise, no actual content. Delete it.

Participants and settings, page 5: 1) The inclusion and exclusion criteria should be more clearly described. For example, college students with a history of mental illness should be excluded. Added. 2) What is meant to ensure sufficient sample size? How is the sufficient sample size calculated? This important content should be written out. 3) How to deal with missing values and quality control? More explanations were added.

The original CD-RISC translation process, page 5: What is the total score? What do high and low scores represent? Added. After the experts provided their opinions, what modifications have you made to the scale? Added in result section, under the translation process. 

Reliability, page 7: Please clearly state the method used for reliability verification. Done.

Response 3:

Thank you for your valuable comments. All of your concerns have been addressed and incorporated in accordance with your suggestions. Please see our revised method section. 

Comment 4:

Results

Demographic information

Table 1: 1) With such a large gender gap, how can the results be representative? 

Response: We appreciate the reviewer's inquiry regarding the gender gap in our study. Our study aims to assess the psychometric properties of the Arabic CD-RISC within the university student context. The observed gender gap in Table 1 stems from our sample's composition, which is centered on university students from selected Jordanian universities. This demographic focus was chosen to provide insights into resilience levels among students in this specific context. We acknowledge this aspect as a limitation and intend to address it in our limitation section. Specifically, the gender gap results from the higher number of female students compared to male students in the Jordanian universities we selected. Additionally, various tests were employed to ensure the representativeness of the sample and its adherence to the normal distribution.

2) Make sure that the percentages in each category add up to 100%. Added. 3) It is recommended to list the scores of each category (Mean±SD) in the table 1. Added were necessary.

Response 4:

Thank you for your valuable comments to enhance my paper. All of your concerns have been addressed and incorporated in accordance with your suggestions. Please see our revised result section. 

Comment 5:

Discussion

As suggested above, first of all, you should conduct a brief descriptive analysis to describe the basic scores and conditions of psychological resilience of Jordanian college students.

Response 5:

Thank you for your valuable comments to enhance my paper. All of your concerns have been addressed and incorporated in accordance with your suggestions. Please see our revised result section. 

Comment 6:

First of all, your research objects are college students from three schools in Jordan. I think this group is not representative, and the article does not explain why college students are chosen for the research. Second, there have been previous studies. Thirdly, your study is only cross-sectional and cannot elaborate the measurement invariance of the scale in depth. Finally the innovation of this study is not enough, and many details were not paid attention to, making the detailed description too brief. 

Thank you and my best,

Response 6: Thank you for your feedback. We address your concerns by clarifying the rationale for selecting university students, highlighting the innovation of our study, and providing a more detailed description of our methodology and findings. We also acknowledge the limitations of our cross-sectional design and consider the potential for future longitudinal research. Your input is invaluable in enhancing the quality of our manuscript. Once again, thank you for taking the time to review our manuscript and provide valuable feedback. Your input will undoubtedly contribute to the overall rigor and impact of our work.

Reviewer #2: Thank you for submitting this manuscript which is both timely and contemporary. Before this manuscript can be considered for publication, there are few points which the author should revisit.

Comment 1:

1. The excluding criteria includes “high school students”. Not sure if this should be included as excluding criterion as the university student cannot have only high school students. Please reconsider.

Response 1:

Thank you for your valuable comments to enhance my paper. Delete it.

Comment 2:

2. Although the sample size of 1220 seems to be adequate, there is no scientific or empirical justification or mentioning of the “rule of thumb” which are widely used in the literature to justify this sample size. Please consult the following key references which may give a better explanation in relation to the number of participants per items:

• Spielberger C, Gorusch R, Lushene R, Vagg P, Jacobs G. State trait anxiety for adults. Mind Garden, Redwood city, Calif, USA. 1983.

• 15.Pedhazur E. Multiple regression in behavioral research, Orlando, FL: Harcourt Brace. 1997.

• 16.Kline RB. Convergence of structural equation modeling and multilevel modeling: na; 2011.

• 17.Field A. Discovering statistics using IBM SPSS statistics: sage; 2013.

Response2:

Thank you for your valuable comments. We have incorporated a new paragraph along with a citation to provide a justified rationale for our sample size. Your input has been instrumental in enhancing the robustness of our study.

Comment 3:

3. Most of the participants were female (i.e., 77%). Does this have an impact on the participants responses?

Response 3:

We appreciate the reviewer's inquiry regarding the gender gap in our study. The observed gender gap in Table 1 stems from our sample's composition, which is centered on university students from selected Jordanian universities. This demographic focus was chosen to provide insights into resilience levels among students in this specific context. We acknowledge this aspect as a limitation and intend to address it in our limitation section. Specifically, the gender gap results from the higher number of female students compared to male students in the Jordanian universities we selected.

Comment 4:

4. The Cronbach alpha for the sample was excellent (i.e. more than 0.9)., but the author should also acknowledge that as the sample size increases, the Cronbach alpha will also increase and this measure become less credible indicator for the internal consistency of the scale.

Response 4:

Thank you for your insightful comment. We appreciate your attention to the Cronbach alpha value in our study. You raise a valid point regarding the potential impact of sample size on the interpretation of Cronbach alpha as a measure of internal consistency. We included this issue in our discussion part. Please see our revised discussion part.

Comment 5:

5. In p. 9. It was mentioned that some changes to the Arabic version of the scale were implemented following the pilot study. Please clarify these changes and why they were implemented?

Response 5:

Thank you for your insightful comment. A new paragraph was added in result under translation headline as follows: The assessment involved a comprehensive evaluation by experts, focusing on the comparative analysis between the translated Arabic version and the original English version of CD-RISC-25. This scrutiny was directed toward ensuring both semantic congruence and cultural resonance. Thus, no items were subjected to deletion in the Arabic version; however, amendments in some words were incorporated into the Arabic version of the items, attuning them to contextual Arabic cultural significance.

Comment 6:

6. It was clear from the scree plot that a two factors model was probably the best, yet the author has opt to use four factors model. Please provide an empirical justification for this decision.

Response 6:

Thank you for your insightful comment. A new paragraph was added in result under construct validity explaining why we opt 4 constructs. A new paragraph was added as follows: As indicated in Figure 1, constructs with eigenvalues greater than 1 are considered significant. In this regard, we identified four constructs with eigenvalues exceeding 1. 

Comment 7:

7. In the limitation section, the author suggest that the “Connor-Davidson Resilience Scale 15 the study focused solely on the Arabic-speaking context, limiting the generalizability to other cultural or linguistic groups “. This is an incorrectly paraphrased sentence. The whole paper is focused on the Arabic version of the scale, so why would the author question the fact that this study was conducted on Arabic-speaking sample?

Response 7:

Thank you for your insightful comment. Delete it .

We hope now that our revised manuscript is acceptable for publication.

---

## [Decision Letter · Decision Letter 1]

25 Sep 2023

PONE-D-23-22861R1Validating the Arabic Version of the Connor-Davidson Resilience Scale Among University StudentsPLOS ONE

Dear Dr. Alfuqaha,

Thank you for submitting your manuscript to PLOS ONE. After careful consideration, we feel that it has merit but does not fully meet PLOS ONE’s publication criteria as it currently stands. Therefore, we invite you to submit a revised version of the manuscript that addresses the points raised during the review process.

The revised paper is imporved, however, you need to meet the reviewers feedback to be accepted. Please ensure that your decision is justified on PLOS ONE’s publication criteria and not, for example, on novelty or perceived impact.

We look forward to receiving your revised manuscript.

Kind regards,

Fadwa Alhalaiqa

Academic Editor

PLOS ONE

Journal Requirements:

Additional Editor Comments :

Dear Author

The revised paper is improved; However, minor changes are required in response to the reviewers' comments.

Reviewers' comments:

Reviewer's Responses to Questions

**Comments to the Author**

1. If the authors have adequately addressed your comments raised in a previous round of review and you feel that this manuscript is now acceptable for publication, you may indicate that here to bypass the “Comments to the Author” section, enter your conflict of interest statement in the “Confidential to Editor” section, and submit your "Accept" recommendation.

Reviewer #1: (No Response)

Reviewer #2: All comments have been addressed

2. Is the manuscript technically sound, and do the data support the conclusions?

Reviewer #1: No

Reviewer #2: Yes

3. Has the statistical analysis been performed appropriately and rigorously? 

Reviewer #1: No

Reviewer #2: Yes

4. Have the authors made all data underlying the findings in their manuscript fully available?

Reviewer #1: Yes

Reviewer #2: Yes

5. Is the manuscript presented in an intelligible fashion and written in standard English?

Reviewer #1: No

Reviewer #2: Yes

6. Review Comments to the Author

Reviewer #1: ID: PONE-D-23-22861R1

Title: Validating the Arabic Version of the Connor-Davidson Resilience Scale Among

University Students

Thank you for providing a chance to review this manuscript.

Detailed information:

Methods and materials

Participants and settings, paragraph 1: Please list separately the software you used and how you analyzed the data. What analysis is AMOS used for?

Results

Demographic information, Table 1: Why is the basic score information of the respondents using this questionnaire not displayed?

Translation process: This is a step before conducting reliability and validity analysis and should not be presented later.

If there are no special requirements from the magazine, all forms should use a three-line form.

Discussion

Do you think the only limitation of your research this time is the uneven gender ratio?

Thank you and my best,

Your reviewer

Reviewer #2: The author has addressed the main points which makes this paper publishable. There are no concerns from my perspectives on the integrity of this paper.

7. PLOS authors have the option to publish the peer review history of their article (what does this mean?). If published, this will include your full peer review and any attached files.

Reviewer #1: No

Reviewer #2: **Yes: **Mansour Mansour. RN.PhD. Associate Professor. Nursing Department. Fatima College for Health Sciences. UAE

---

## [Author Response · Author response to Decision Letter 1]

26 Sep 2023

Response to Journal and Reviewers comment

Journal requirements

Response: 

We have edited all references as per your request. However, there is one reference that is important to include, as it measures and validates CD-RISC among Thai people. We have corrected and cited it properly as follows:

24. McGillivray K, editor Ho R. Validation of the Connor Davidson resilience scale (Cd-Risc) as applied within the Thai context. Scholar: Human Sciences. 2017;8(2):178-187. Retrieved from http://www.assumptionjournal.au.edu/index.php/Scholar/article/view/2508

Dear editor,

We would like to thank you and the reviewer for their valuable comments and feedbacks. Point-by-point responses to reviewers are listed below.

Editor comments:

The revised paper is improved; however, you need to meet the reviewers feedback to be accepted. 

Response: Thank you for your valuable feedback. As you will notice, we respond to each reviewer's feedback, as illustrated below.

Reviewer comments

Reviewer #1: 

ID: PONE-D-23-22861

Title: Validating the Arabic Version of the Connor-Davidson Resilience Scale Among University Students

Thank you for providing a chance to review this manuscript.

Detailed information:

Comment 1:

Methods and materials

Participants and settings, paragraph 1: Please list separately the software you used and how you analyzed the data. What analysis is AMOS used for?

Response 1:

Thank you for your valuable comments. A new paragraph was added as follows: 

“In the data analysis phase, this research relies upon the utilization of two distinct software programs, namely SPSS version 23 and Analysis of Moment Structures (AMOS) software version 25. SPSS is employed for the computation of fundamental statistical measures such as frequencies, means, standard deviations, explanatory factor analysis (EFA), and the Cronbach's alpha coefficient. On the other hand, the AMOS software is employed to conduct a comprehensive examination of the interrelationships among items, particularly during the confirmatory factor analysis (CFA) stage.

Comment 2:

Results

Demographic information, Table 1: Why is the basic score information of the respondents using this questionnaire not displayed?

Response. I modify the title of Table 1 as “Demographic information and resilience score”, and at the end of Table 1, I display the score of resilience (dimensions and the overall score) among participating students in this study. 

Translation process: This is a step before conducting reliability and validity analysis and should not be presented later.

Response: Please see our revised manuscript. The translation process in method section and result section is presented before conducting reliability and validity analysis.

If there are no special requirements from the magazine, all forms should use a three-line form.

Response: Done

Comment 3:

Discussion

Do you think the only limitation of your research this time is the uneven gender ratio?

Response 3:

Thank you for your valuable comment. I added more limitations to my study, please see our revised manuscript. 

Thank you and my best,

Your reviewer

Reviewer #2: The author has addressed the main points which makes this paper publishable. There are no concerns from my perspectives on the integrity of this paper.

Response: Thank you for your valuable feedback. 

We hope now that our revised manuscript is acceptable for publication.

---

## [Editor Report · Decision Letter 2]

12 Oct 2023

Validating the Arabic Version of the Connor-Davidson Resilience Scale Among University Students

PONE-D-23-22861R2

Dear Dr. Alfuqaha,

We’re pleased to inform you that your manuscript has been judged scientifically suitable for publication and will be formally accepted for publication once it meets all outstanding technical requirements.

Kind regards,

Fadwa Alhalaiqa

Academic Editor

PLOS ONE
---

## [Editor Report · Acceptance letter]

16 Oct 2023

PONE-D-23-22861R2 

Validating the Arabic Version of the Connor-Davidson Resilience Scale Among University Students 

Dear Dr. Alfuqaha:

I'm pleased to inform you that your manuscript has been deemed suitable for publication in PLOS ONE. Congratulations! Your manuscript is now with our production department. 

Kind regards, 

on behalf of

Pro Fadwa Alhalaiqa 

Academic Editor

PLOS ONE